# Glomerales Dominate Arbuscular Mycorrhizal Fungal Communities Associated with Spontaneous Plants in Phosphate-Rich Soils of Former Rock Phosphate Mining Sites

**DOI:** 10.3390/microorganisms10122406

**Published:** 2022-12-05

**Authors:** Amandine Ducousso-Détrez, Robin Raveau, Joël Fontaine, Mohamed Hijri, Anissa Lounès-Hadj Sahraoui

**Affiliations:** 1Unité de Chimie Environnementale et Interactions sur le Vivant (UCEIV), Université du Littoral Côte d’Opale, UR 4492, SFR Condorcet FR CNRS 3417, CEDEX, 62228 Calais, France; 2Institut de Recherche en Biologie Végétale (IRBV), 3 AgroBioSciences, Université de Montréal, Montréal, QC H1X 2B2, Canada; 3INRAE, UMR SAVE, Bordeaux Science Agro, ISVV, 33882 Villenave d’Ornon, France; 4African Genome Center, Mohammed VI Polytechnic University (UM6P), Ben Guerir 43150, Morocco

**Keywords:** arbuscular mycorrhizal fungi, phosphorus, rock phosphate, diversity, taxonomic composition

## Abstract

Arbuscular mycorrhizal fungi (AMF) are key drivers of soil functioning. They interact with multiple soil parameters, notably, phosphorus (P). In this work, AMF communities of native plants grown spontaneously on former mining sites either enriched (P sites) or not enriched with P (nP sites) by mining cuttings of rock phosphate (RP) were studied. No significant differences were observed in the root mycorrhizal rates of the plants when comparing P and nP sites. The assessment of AMF diversity and community structure using Illumina MiSeq metabarcoding and targeting 18S rDNA in roots and rhizospheric soils showed a total of 318 Amplicon Sequence Variants (ASVs) of Glomeromycota phylum. No significant difference in the diversity was found between P and nP sites. Glomeraceae species were largely dominant, formed a fungal core of 26 ASVs, and were persistent and abundant in all sites. In the P soils, eight ASVs were identified by indicator species analysis. A trend towards an increase in Diversisporaceae and Claroideoglomeraceae and a reduction in Paraglomeraceae and Glomeraceae were noticed. These results provide new insights into AMF ecology in former RP mining sites; they document that P concentration is a driver of AMF community structures in soils enriched in RP long term but also suggest an influence of land disturbance, ecosystem self-restoration, and AMF life history strategies as drivers of AMF community profiles.

## 1. Introduction

Many efforts have been deployed to sustain agricultural production systems while reducing their environmental footprint. Ecosystemic contributions of microbial resources to improve and maintain soil fertility are among the strategies that have been utilized. In particular, arbuscular mycorrhizal fungi (AMF) have gained a large interest as biological input in agroecosystems [1,2,3]. Indeed, AMF are ubiquitous fungi that form symbiotic associations with most plant species, including important crop plants, such as vegetables and cereals [4,5]. They are obligate biotrophs, receiving their carbon sources in the form of photosynthetic sugars and fatty acids from plant hosts [6,7]. In exchange, elements such as phosphorus (P), nitrogen, sulfur, or microelements, such as copper and zinc, may also be transferred to the host via arbuscules, favoring plant nutrient supply [1,8,9]. Mycorrhizal symbiosis can also contribute to plant protection against biotic or abiotic stresses [10,11,12,13]. In addition, evidence also indicate that AMF can be critical components to improve soil structure, water retention capacity, and soil fertility and reduce soil nutrient leaching [1,14,15,16,17]. Concomitantly, the multiple benefits that AMF confer to their hosts and agroecosystems’ functioning have raised opportunities for their application as commercial biofertilizers. In particular, AMF increase the uptake of relatively immobile ions (such as phosphate ions) for their host plant due to an extensive extraradical hyphal network that extends beyond the zone of direct uptake by roots. In this way, they contribute to greater soil prospection and enable mycorrhizal plants to access more soil P resources. Therefore, AMF have great importance to productivity and plant growth in most ecosystems [1,18]. However, despite the great potential of AMF and the evidence of their relevance for agriculture [19], a deeper understanding of AMF ecology is of primary importance to improving engineering and the use of AMF-based bioinoculants in ecosystems.

In such context, significant improvements have been made in the detailed analysis of AMF communities among complex habitats using AMF gene markers and high-throughput sequencing technology for taxonomic identification, phylogenetic reconstruction, or quantification of AMF [20,21,22,23]. Thus, due to such improvements, much compelling evidence has reported that different abiotic and biotic factors interfere with AMF communities at either a global or local scale [24,25,26,27]. In particular, the host plant can be considered a strong factor in shaping AMF communities [28,29,30], but these are also known to be sensitive to interactions with other soil organisms [31,32]. Edaphic properties [24,29,33] and climatic parameters [10,25,34,35] are also recognized as strong drivers of AMF communities. In addition, some studies have suggested a high diversity of AMF communities in natural habitats [36,37] while a lower diversity tends to be observed in agricultural ecosystems [38]. Notably, AMF communities’ characteristics may vary with land use (usage intensity, cultivation duration, pasture, etc.) and with agronomic practices for soil management (chemical inputs, plowing, tillage, crop rotation, land conversion, etc.) [39,40]. In particular, the responses of AMF to soil P were largely investigated [41,42,43]. So, research efforts on the understanding of P as a key driver of AMF colonization, diversity, and community composition have gained interest, in particular, for identifying guides in the development of AMF inoculants and their use with a reduction in the use of chemical fertilizers [44,45]. In particular, how AMF communities interact with rock phosphate (RP) has attracted considerable attention because RP is a natural, low industrial-cost phosphate input that has more ecological and sustainable acceptability than chemical fertilizers. However, P as a key player in shaping AMF communities still remains a controversial subject in the literature. Thus, the impact of P concentrations on AMF colonization as well as on the diversity and composition of AMF communities is regularly discussed [28,39].

Thus, the current work aims at characterizing the diversity and composition of AMF communities across a former mining area of RP where two types of sites occurred: sites enriched or not enriched with phosphate deposits originating from the mining activity. The AMF community profiles in the soil and root samples were determined using MiSeq amplicon sequencing targeting 18S rDNA that generates Amplicon Sequence Variants (ASVs). The diversity and composition of AMF were analyzed at several levels of taxonomic resolution, and the potential role of the P gradient and soil restoration process in AMF community composition was discussed.

## 2. Materials and Methods

### 2.1. Study Area and Experimental Design

This work was carried out in the region of the “Phosphatières du Quercy” (southern France; N 44.351827, E 1.691021). This area is characterized by numerous paleokarst whose fillings are enriched in phosphorite (i.e., a phosphatic ore containing variable proportions of tricalcium phosphate). There, in the past, mining extraction was carried out in the form of open-pit mines (Appendix A), and some excavated spoils were abandoned near the extraction points. Thus, each selected site corresponds to a former mining point (i.e., a site without enrichment with RP ores, referred hereafter as “nP site”) associated with its spoil area (i.e., a site with ore deposits due to mining exploitation process, referred to as “P site”). Today, in this former mining area, the dominant landscapes are dry grasslands that are punctually wooded (Appendix A).

The P concentrations are known to be more or less high in each karst phosphatic fill [46]. Three nP were therefore chosen, expecting significantly high P concentrations. The choice of locations was also constrained (i) by the possibility of locating potential P and nP sites in close proximity to each other from the ground surface and (ii) by the necessary presence of the same set of species constituting the plant sampling. Thus, four native herbaceous mycotrophic species were sampled: *Ranunculus bulbosus*, *Bromus sterilis*, *Taraxacum officinale,* and *Dactylis glomerata*, and three plants per species were collected. From these 3 plants, the soil closely attached to roots (i.e., rhizospheric soil) was manually removed by gentle agitation and pooled into a single soil fraction. Then, the roots were washed with sterile water to get rid of remaining soil particles and then pooled into one root fraction per species (Figure 1). Soil and roots were then, respectively, kept at 4 °C and −20 °C until further DNA extraction. This was repeated for each plant species in each site, leading to 24 root samples (6 sites × 4 plant species) and 24 rhizospheric soil samples. Detailed soil properties for each site are available in Appendix A.

### 2.2. Estimation of Arbuscular Mycorrhizal Root Colonization

Immediately after plant harvesting, from each plant species and site, fractions of fresh roots were stained using nonvital Trypan blue (0.5 g·L^−1^) according to the method described by [47]. From each stained root batches (24 batches), fragments of 1 cm were sampled, and 3 intersections per root fragment were examined through microscopic observations (Nikon Eclipse E600, ×100 magnification). The rate of root colonization by AMF structures (vesicles and arbuscules) was quantified as described by [48].

### 2.3. Soil Physicochemical Properties

In each site, the soil physicochemical properties (pH, soil texture, chemical composition) were measured by the CIRAD-US Analyse laboratory (Montpellier, France) using inductively coupled plasma spectrometry, atomic emission spectrometry, and X-ray fluorometry. The 6 soils were characterized by near-neutral pH, ranging between 6.9 and 7.3. Regarding granulometry, all topsoil (0–30 cm) resulted in high levels of clay with additionally higher percentages of coarse sands in P soils. In nP soils, total P concentrations ranged from 1057.1 to 1496.3 mg·kg^−1^ while in P soils, they ranged from 2880.0 to 13,927.9 mg·kg^−1^. Likewise, available Olsen P concentrations in P soils were between 46.1 mg·kg^−1^ to 339.5 mg·kg^−1^ while nP soils were characterized by values ranging from 5.04 to 12.82 mg·kg^−1^. Thus, by comparing matched soils from the same localization, P and Olsen P ratios ranging from 2.07 to 13.05 and from 9.14 to 41.96 Pi, respectively, were obtained (Appendix A), which allowed to qualify the P sites as high-P sites, and the nP sites as low-P sites.

### 2.4. DNA Extraction, PCR, and Sequencing

#### 2.4.1. DNA Extraction

Genomic DNA extraction from the rhizospheric soil samples was directly performed from 250 mg of soil using a Nucleospin Soil^®^ kit (Macherey-Nagel, Düren, Germany) according to the manufacturer’s instructions.

Genomic DNA extraction from root samples was operated from roots initially frozen in liquid nitrogen and then reduced to fine powder. Then, the extraction from 250 mg of ground roots was performed using Cetyltrimethylammonium bromide (CTAB, 1.4 M NaCl_2_ 100 mM Tris-HCl, pH 8.0; 20 mM EDTA, pH 8.0; 2% CTAB), Polyvinylpyrrolidone (PVP 1% *w*/*v*), ß-Mercaptoethanol (5% *v*/*v*), and activated charcoal (0.5% *w*/*v*) extraction (30 min; 55 °C). A centrifugation step was then carried out (10 min; 16,000× *g*), and lysate extraction was performed with chloroform: isoamyl alcohol (24:1). DNA precipitation was obtained in isopropanol (1 h incubation; 25 °C) followed by centrifugation (10 min; 700× *g*). After three washings (with ice-cold ethanol (70%) followed by centrifugation—10 min at 900× *g*) and air-drying at room temperature (approx. 90 min; 20 °C), the DNA pellet was dissolved in 50 µL of TE buffer (10 mM Tris-HCl, pH 8.0; 1.0 mM EDTA, pH 8.0) [13,49,50].

All extractions were performed in triplicate. DNA quality of root and soil DNA extracts was assessed using 1% (*w*/*v*) agarose gel, and measures of the 260/280 nm and 260/230 nm ratios were performed with a SpectraMax^®^ iD3 device (Molecular Devices LLC, Sunnyvale, CA, USA). After measures of DNA concentrations, root and soil DNA extracts were normalized to 25 ng·L^−1^ for further analyses. The extracted DNA was stored at −20 °C until further use.

#### 2.4.2. PCR Targeting the 18S rRNA Gene of AMF

The DNA of AMF was specifically targeted using Nested-PCR (Surecycler 8800, Agilent Technologies, Les Ulis, France) with a set of two primer pairs as previously published by [50]. Thus, the first-round PCR was performed using the AMF-discriminating primer pair AML1 (3′-ATCAACTTTCGATGGTAGGATAGA-5′) and AML2 (3′-GAACCCAAACACTTTGGTTTCC-5′) which generate amplicons of the small 18S subunit of the rRNA gene of about 800 bp in length [51,52]. The PCR conditions were as follows: initial denaturation at 94 °C for 3 min followed by 35 cycles at 94 °C for 1 min, 45 °C for 1 min, and 72 °C for 1 min, and a final elongation step at 72 °C for 5 min [13]. PCR reactions were performed in a reaction volume of 25 µL, and reagents were as follows: 5 µL of Q5 (5X) reaction buffer, 0.25 µL of Q5^®^ High-Fidelity DNA Polymerase (New England Biolabs France, Evry, France), 0.8 µL of each primer (0.4 µM), 1 µL of dNTPs (0.2 mM), 1 µL of DMSO, 1 µL of BSA (100 µg·mL^−1^), and 1 ng of DNA template [13,51].

From the 800 bp length amplicons obtained after the first-round PCR, which are not compatible with Illumina sequencing technology, nested PCRs were performed using an in-house set of internal primers nu-SSU-0595-5-F (ACACTGACGACATGGTTCTACA CGGTAATTCCAGCTCCAATAG) and nu-SSU-0948-3-R (TACGGTAGCAGAGACTTGGTCT TTGATTAATGAAAACATCCTTGGC). This set of primers flanked the V4 region of 18S and hence downsized the length of the amplicons to about 400 bp, which is compatible with MiSeq 300 paired-end sequencing [53]. The two primers were complemented with CS1 and CS2 barcoded adapters [13,53]. The reaction mixture was the same as previously described, but 1 ng of DNA from the first-round PCR as template was used. The reaction conditions for the second PCR were as follows: initial denaturation at 94 °C for 3 min followed by thirty-five cycles at 94 °C (1 min), 58 °C (1 min), and 72 °C (1 min), and a final elongation step at 72 °C (5 min).

#### 2.4.3. Sequencing

From genomic DNA of root and soil samples collected for each plant species harvested in each site, PCR reactions were performed in triplicates for each sample, and these PCR products were then pooled together. Sequencing was processed by Genome Quebec Innovation Centre (Montreal, QC, Canada) by using an Illumina MiSeq platform generating 2 × 300 bp paired-end reads.

#### 2.4.4. Nucleotide Sequence Accession Number

The 18S rRNA gene sequences of the raw data set have been deposited in the NCBI Sequence Read Archive database under the project accession number PRJNA877825.

### 2.5. Bioinformatic and Statistical Analyses

Bioinformatic process and statistical analyses were operated in the R 4.0.2 software (R Core Team, 2019) environment. The DADA2 pipeline (v. 1.16) [54], an open-source program implemented in R package (https://benjjneb.github.io/dada2/tutorial.html accessed on 1 October 2022), was used to process the sequencing data. Briefly, sequence reads were filtered and trimmed using optimized parameter settings as recommended. Sequence reads were dereplicated, denoised, and merged using DADA2 default parameters. Then, sequences were aligned and categorized to infer amplicon sequence variants (ASVs) grouping amplicon sequencing data by using 100% of sequence identity [55]. Sequences presenting only once (singletons) in the whole data set were eliminated. Validity of sequencing depth was controlled from rarefaction curves, computed using the “rarecurve” function from the Vegan package in R.

The taxonomic assignment of ASVs was performed following a previously established two-step approach [13,53]. In the first step, the Silva v132 database formatted for DADA2 was used to assign ASVs from kingdom to genus (minimum bootstrap 80) with the assignTaxonomy () command and the blast parameters as constitutively formatted for DADA2 [56]. From the ASVs assigned as AMF with Silva database, affiliation of each ASV was assessed at the genus level by performing a BLAST analysis against the NCBI database and MARJAAM, a web-based database containing referenced Glomeromycota DNA sequence data. The ASVs identified as non-Glomeromycota at the phylum level at the end of these two stages of assignment were excluded from further analyses.

In a second step, aiming at a refined taxonomic identification of each ASV, a phylogenetic tree was subsequently constructed as proposed by Stefani et al., 2020 [53]: multiple alignment including ASVs identified as Glomeromycota and multiple consensus sequences was computed using the web portal Kalign [57] (http://msa.sbc.su.se/ accessed on 1 October 2022). Then, a maximum-likelihood tree was obtained using RAxML v8.2.10 [58], through the CIPRES web portal [59]. Visualization of the output was finally obtained with the FigTree v1.4.4 program.

With the aim to identify the influence of former RP inputs on AMF communities, we studied AMF communities across the six sampling sites as well as inside root or rhizospheric soil using proxy richness, alpha diversity, distribution, and abundance of taxa. Then, data relative to P sites were compared to those of nP sites, inside each location, or after pooling data from the 3 P sites on the one hand and the 3 nP sites on the other hand. The Chao1 richness estimator and alpha diversity indices (Shannon and Simpson) [60,61,62] were computed from the plot_richness () function using the phyloseq R package.

The normality and homoscedasticity of the data were assessed from the Shapiro–Wilk and Bartlett tests, respectively. If both conditions were verified, ANOVA analysis complemented with a posthoc Dunn test was carried out. Otherwise, a Kruskal–Wallis nonparametric test (“kruskal.test” function in R) was used. The significance of the statistical analyses was considered for α = 0.05.

Venn diagrams were constructed with the Vegan package to visualize the numbers of ASVs that are either specific to one site or shared between paired sites inside the mining area; significance of the difference in ASVs numbers was then conducted using chi2 test. Indicator species analysis was performed using the multipatt function from indicspecies package [63] to compute the indicator value index (IndVal), i.e., a measure of specificity (based on abundance values) and fidelity (computed from presence data) of each ASV to a targeted clustering group of samples related to a targeted ecological condition [60]. This index, therefore, led to identifying ASVs (hereafter referred to as “indicator ASVs”) that can be considered closely related to the ecological condition of their group [64]. The significance of IndVal of each species was assessed by a random permutation procedure from 9999 permutations with a significance at the α = 0.05 level (IndVal’s *p*_values with the p.adjust () function).

## 3. Results

### 3.1. Mycorrhizal Colonization of Plant Roots

The AMF root colonization was estimated in each sampling site from 135 microscopic observations of different individuals per plant species. Root colonization rates ranged from 13 to 63% across the different sampling sites (L1-P vs. -nP, L2-P vs. -nP, and L3-P vs. -nP) and plant species (*Ranunculus bulbosus*, *Bromus sterilis*, *Taraxacum officinale,* or *Dactylis glomerata*). However, when considering all individuals of the four plant species as a pool (to reduce a putative effect only linked to plant species), no significant difference in mycorrhizal colonization between plants growing in P versus nP sites was observed, whether the estimate was made for the entire mining area or within each individual localization. Thus, considering the entire mining area, the mean percentages reached 42 and 43% in plant roots sampled in P versus nP sites, respectively (Table 1).

### 3.2. Analysis of AMF Diversity

After Illumina Miseq sequencing, the global dataset resulted in a total of 4,795,981 raw MiSeq reads (yielded across 24 soil samples and 24 root samples, Appendix A). After quality filtering and chimera removal, a total of 3,527,278 sequences were retained as the AMF 18S rDNA dataset (Appendix A). The rarefaction curves obtained after quality filtering reached a plateau, indicating we had a good representation of the microbial community, as most of the abundant species were represented and the sequencing depth effort was adequate to progress to further AMF community analysis (Appendix A).

From the AMF 18S rDNA dataset and using the Silva v132 database formatted for DADA2, 392 ASVs were inferred and identified as Glomeromycota. Among them, 74 ASVs were not assigned at the phylum level after inferring the phylogenetic tree (Appendix A). These ASVs were therefore excluded from further analysis. Thus, a total of 318 ASVs were kept for further comparisons of the AMF communities within P and nP soils.

### 3.3. Richness and Alpha-Diversity

The diversity (Shannon and Simpson indexes) and richness (Chao1 estimator) were used to compare AMF communities across P versus nP sites as well as between root or rhizosphere biotopes. A significant difference was observed between the two habitats, with higher diversity in roots compared to rhizosphere soil biotopes. In contrast, no significant difference was recorded comparing the P and nP sites either for the entire mining area (Figure 2) or within each localization (data not shown). No impact of P versus nP status was noticed in either the root or rhizosphere biotopes (data not shown).

### 3.4. Taxonomic Assignment of ASVs

The 318 ASVs identified as Glomeromycota were assigned to Glomerales (81%), Diversisporales (12%), Paraglomerales (3%), and Archaeosporales (2%) (Figure 3A) while no taxonomic assignment at order level was obtained for 2% of them. Among them, five families were identified: Glomeraceae, Claroideoglomeraceae, Diversisporaceae, Paraglomeraceae, and Archeosporaceae. About 20.7% of the ASVs could be affiliated to a genus, mainly identified as *Rhizophagus*, *Funneliformis*, and *Glomus* (18, 15, and 14 ASVs, respectively), but also, to a lesser extent, as *Archaeospora*, *Paraglomus*, *Septoglomus*, and *Claroideoglomus* (Appendix A). With assignment up to the species rank, 39 ASVs (12.2%) could be named in line with referenced strains in the published databases. They were representative of eight species and six genera, namely *Archaeospora trappei, Claroideoglomus lamellosum, Funneliformis mosseae, Glomus indicum*, *G. iranicum*, *Rhizophagus irregularis*, *R. vesiculiferus*, and *Septoglomus africanum* (Figure 3B). Among them, the prevalent ones in terms of both read count and ASV numbers were *F. mosseae* (10 ASVs totaling together 12,591 reads, i.e., 2.7% of the readings), *G. indicum* (9 ASVs with 2698 reads, i.e., 0.6%), and *R. vesiculiferus* (6 ASVs with 8033 reads, i.e., 1.75%). Among ASVs assigned up to the species level, ASV 36, referred to as *F. mosseae*, was dominant in terms of relative abundance (2.6%) (Appendix A).

Regarding variations across the samples (Figure 3C), Glomeraceae were largely dominant in all samples (i.e., across root and soil samples for each site) with Diversisporaceae and Paraglomeraceae mainly observed in soil.

### 3.5. ASVs Shared by the Six Mining Sites

When having a look at the different experimental conditions, a core of 26 ASVs (about 8.2% of the total ASV number) shared by the six sampling sites was identified (Appendix A). These ASVs, ubiquitously present, correspond to about 64% of the total read number retained after the bioinformatic processing, taxonomic, and phylogenetic assignment. Among them, three were dominant (ASV61, ASV62, and ASV208), contributing to 57.1% of the read number of the 26 ASVs. With the exception of two ASVs assigned to Diversisporaceae, all core ASVs belonged to Glomeraceae. Two of them were identified up to the species level with a taxonomic identity inferred to *R. vesiculiferus*.

### 3.6. Dissimilarity in the AMF Community Composition and Identification of Indicator Species Comparing P and nP Sites

By comparing P versus nP sites, ASVs that were specific (i.e., occurring exclusively across the P versus nP sites) and ASVs that were shared (i.e., observed at both P and nP sites) were observed, as shown in Figure 4. Thus, considering the complete mining area, we identified 100 ASVs that are shared by the two profiles, representing 31.4% of the total ASV number but 86.5% of the total read number. In contrast, 138 ASVs (43.4%), accumulating 8.6% of reads, were identified as nP-site-specific (hereafter referred to as “nP-specific ASVs”). Concomitantly, 80 ASVs (25.2%) were P specific, displaying 4.85% of the reads (Figure 4).

Considering each localization, the same trend was observed, with P sites showing a lower number of specific ASVs compared to non-P sites. The most contrasting difference was observed in location two, where the number of nP-specific ASVs was almost twice the amount of P-specific ASVs (52 ASVs, or 41.6%, versus 28 ASVs, or 22.4%, respectively) (Figure 4). Variations in ASVs abundance among P vs. nP sites were also observed, as depicted in Figure 5, focusing on the 50 most abundant ASVs. In addition, it highlights the higher abundance of ASV 61, 62, and 208 in all sites.

Moreover, indicator species analysis was performed to search ASVs indicative of the following sample groups: the P versus nP group, the root versus soil group, as an individual group, or in combination (Appendix A). Thus, in particular, seven nP-indicator ASVs and one P-indicator ASV were identified. All of them were assigned to Glomeraceae.

Regarding the taxonomic classification, no Archaeosporaceae member was observed in P soils. Paraglomeraceae and Glomeraceae tend to be reduced in P soils in terms of ASV percentages (Figure 6). In contrast, Diversisporaceae and Claroideoglomeraceae families tended to have increased with the P content. Lower percentages of assigned ASV were recorded at genus and species levels in P soils (Figure 6).

In addition, comparisons between the three P sites were made. Different ASVs between the three P sites were identified with a higher number of site-specific ASVs and few shared ASVs, as shown in the Venn diagrams (Appendix A). The same trend occurred comparing the nP sites (Appendix A).

## 4. Discussion

In this study, we investigated AMF root colonization and taxonomic profiles of AMF communities in a former mining area comparing P and nP sites. These sites were characterized by total P concentrations ranging from 1067.1 to 13,927.9 mg·kg^−1^ while the values conventionally reported in the literature generally range from 40 to 3000 mg·kg^−1^ [65,66,67]. Additionally, the soluble P (P Olsen) concentrations ranged from 5 to 339.5 mg·kg^−1^, which is largely above those displayed, for instance, in soils supplemented with mineral or organic fertilizers [67]. Such high concentrations in soluble P content are scarce and unexpected. Indeed, in the phosphatieres, soils are enriched with apatite, a form of highly insoluble soil P, and aluminous phosphate parageneses, clays, and sands [46]. As the soluble orthophosphate ions act as chemical ligands for soil compounds, P is generally considered to be highly unavailable in such soils and measured soil pH values (neutral). It is therefore appropriate to question the putative origin of such soluble P concentrations. Our work on the bacterial and fungal communities of the same sites revealed the presence of numerous taxa known to host phosphate-solubilizing microorganisms, and phosphate-solubilizing bacteria have also been successfully isolated (data not shown). These microorganisms may be factors of P release from its unavailable form. However, this would imply that there is no feedback to regulate microbial P solubilizing metabolic activity in the presence of high phosphate substrate concentrations. However, to the best of our knowledge, there is no data in the literature on this point. In any case, such variations in P content provided the opportunity to examine the potential role of P content as a driver of AMF communities.

### 4.1. High P Contents in the Mining Area do Not Impact AMF Root Colonization

In this study, no difference in AMF colonization rates was observed when comparing P to nP soils, which therefore excluded a negative impact of high P levels on AMF colonization in the reclaimed soils studied.

Yet, a decrease in mycorrhizal colonization when P availability is high has been described by numerous authors [68,69,70,71,72,73,74,75]. Additionally, decreases in the abundance and richness of AMF communities were also observed in soil and roots after P fertilization [76,77,78,79]. However, some authors underlined that the decreased AMF colonization upon P fertilization was not systematic [80,81,82,83,84]. Moreover, a moderate amount of P fertilizer could improve AMF diversity while higher amounts of P reduce it [85,86], suggesting that the AMF response to soil P could be related to the P concentration in the soil [87]. On the contrary, Higo et al. [39] observed that the P fertilizer level did not impact either the AMF root colonization, diversity, or community structure.

Such data, therefore, highlight that the AMF response to P is a complex process and underline that the P level is not a reliable predictor of AMF occurrence, root colonization, or community patterns [88,89]. In particular, the AMF response to P may be strongly impacted by sampling time [90], agricultural management [39], or host plant species [28,42,83]. For instance, Tang et al., 2016 [83] showed that for faba bean, an excess of P fertilization did not significantly affect the mycorrhizal rates when compared to a low P and no P treatment. Conversely, the durum wheat root colonization was significantly lower at a high P level, and the arbuscular rate drastically decreased at both low and high P fertilization compared with the unfertilized treatment. In this line, we question the role of the site mining history possibly acting as a stronger driver of AMF communities compared to the P level.

### 4.2. AMF Community Characteristics across Sites: Consequences of Site Mining History Rather Than P Concentrations

In this report, we observed a large taxonomic similarity between AMF communities at the family level whether the soils were enriched in P or not. Glomeraceae was largely prevalent throughout the entire mining ecosystem whereas Archeosporaceae, Diversisporaceae, Paraglomeraceae, and Claroideoglomeraceae were under-represented. Paraglomeraceae and Gigasporaceae were not detected. In addition, we highlighted 26 ASVs shared by all sampling sites, three of which were very abundant.

Several hypotheses may be put forward to explain the low richness or diversity in terms of taxa at the family level: either the absence or under-representation of some AMF families and in contrast, the over-representation of Glomeraceae.

Firstly, Glomeraceae is classically recognized as the most widespread family in global, natural, and managed ecosystems [91,92]. In addition, anthropogenic activities are known to have considerable influence on AMF communities. Notably, intensive agricultural practices, such as soil disturbance by tillage [93] or P fertilization, can adversely affect AMF colonization and diversity [75,88,94,95]. Notably, some *Glomus* spp. are commonly found in soils subjected to fertilization and disturbance whereas others, especially *Scutellospora* spp., are indicative of minimally disturbed soils [96]. In the same way, some results interestingly suggest that soil AMF communities may differ according to land use history and time after land use conversion, suggesting that their diversity and composition may be a legacy of ancestral adaptations to historical habitat and exhibit niche conservatism [33,96]. It has also been suggested, that intensively managed agroecosystems are greatly simplified compared to natural ecosystems. For instance, comparisons between AMF communities in agrosystems managed in different ways showed that families such as Gigasporaceae or Acaulosporaceae are more frequently reduced or eliminated than Glomeraceae [97,98].

In theory, the AMF’s life-history strategy (LHS) has been proposed to explain the dynamics, variations, and prevalence of some taxa among AMF communities [99,100]. This concept identifies K-strategists (i.e., AMF competitors that evolved traits to enhance survival in stable environments where competition is high) and r-strategists, or ruderals, that evolved traits to survive under stressful conditions and in disturbed environments, investing their energy mainly in the production of numerous offspring. Notably, Glomeraceae may show an opportunistic behavior similar to r-strategists [101,102,103]. Thus, taxa resilience and the prevalence of Glomerales in disturbed environments could be explained through abilities such as their higher sporulation, ability to produce high numbers of new hyphae from injured hyphae after mechanical disruptions with healing mechanisms, and anastomosis processes [103,104,105]. These properties are probably more suited to perturbed environments, and may enable them to spread and reestablish more quickly [106,107]. In addition, it has been shown that representatives of the *Glomus* genus promptly respond to carbon resource shortages with direct investment in the production of spores for reproduction, as is expected for r-strategists [108]. Conversely, Gigasporaceae may resemble the LHS of K-strategists; they mainly spread or colonize through spores from an intact mycelium within a long life cycle [109], which does not facilitate recovery and resilience from disturbance.

With regard to these theoretical assumptions, homogeneity across AMF communities as reported in our study might be explained partially by the landscape history, i.e., firstly by the drastic soil perturbation due to mining activities and secondly, by the restoration and revegetalization of soil during more than one century. Thus, we hypothesize that the soil recolonization by AMF would have taken place with the extensive development of AMF species functionally adapted to rapid recolonization and the stress of high P concentrations. Here, Glomeraceae could be relevant candidates while other AMF families could have been drastically or progressively suppressed, becoming low or nondetectable today under the influence of persistent high P concentrations in the soil for more than a century. In particular, no representative of Gigasporaceae has been identified in the sampled soils. Yet, it has been shown that Gigasporaceae provide functional complementarity to other AMF families in P uptake [110]. In our conditions, due to high soluble P concentrations in the sampling soils, we hypothesize that additional benefits might not have been supplied to the host by such taxa, resulting in their absence. Indeed, the assimilation of P by AMF-colonized plants reflects the sum of P uptake directly by plant cells and indirectly via the AMF pathway [69]. This, however, becomes subsidiary to plants when the P availability is high in the soil matrix and when direct plant uptake is sufficient to provide enough Pi without the help of the indirect mycorrhizal pathway. Consequently, AMF may no longer be essential for P uptake while their other functions (contribution to the nutrient or water supply as well as stress resistance) would remain major, hence, the persistent presence of AMF in P-rich soils. Therefore, we also hypothesize that, besides the predominance of Glomeraceae, the decline or loss of some AMF taxa, and the decrease in overall diversity and afferent mycorrhizal functioning, Glomeraceae recruited by plants may display a large functional diversity and complementation to gain the maximal benefits from the AMF communities in favor of the soil and host plant and for maintaining ecosystem functions.

In line, the occurrence of a representative of *Funneliformis* described in the literature as a ruderal and stress indicator taxon [100] and as a generalist AMF [111] may be compatible with the disturbed land story of sampling sites.

Following our results, the meaning and role filled by the set of persistent ASVs capable of coping with radically contrasting and altered environments can also be questioned. One could hypothetically suggest that this microbial core could correspond to taxa with basic functions (such as nutrient transfer, mainly C and N) and functional features (not yet identified) that confer some level of robustness, resistance, resilience, or functional redundancy in the face of high P content, environmental disturbances, and progressive soil restoration over time [112,113,114]. However, data about the taxonomic affiliation of the core members are insufficient to go further about their functionalities. Moreover, data in the literature allowing functional inference to a given AMF species are still limited since the taxonomy and identification of AMF species are still to be refined.

### 4.3. A slight Impact of Soil P on AMF Assemblages Is Resolved at ASV Level

In this study, we identified shared and specific ASVs characterizing the P and nP sites; on the other hand, a lower richness and abundance of AMF communities were observed for the P profile compared to the nP profile. This was particularly pronounced in one site (the S3 site) which was furthermore characterized by the highest concentrations of total and soluble P.

Moreover, through an indicator species analysis, we statistically identified ASVs that are indicative of P and nP sites based on the indicator value calculation as a measure of specificity and fidelity of each ASV to the targeted clustering groups, i.e., P vs. nP site group [63]. Notably, both P- and nP-indicator ASVs were identified but with a lower number of P-indicator ASVs. As a whole, these results clearly differentiate the P and nP sites and therefore highlighted that the AMF community is responsive to P as an edaphic factor in driving AMF communities when analysis is performed with a stringent level of taxonomic resolution.

## 5. Conclusions

In this study, we identified a large similarity of AMF community compositions at the family rank as well as the persistence of a set of ASVs along the drastic P gradient inside the mining area sampled. This leads to the conclusion that AMF community profiles were robust across the contrasting soil P levels. Nevertheless, the shared and specific ASVs from the P vs. nP sites were also identified as well as indicator ASVs of the P vs. nP sites which, therefore, differentiated the P vs. nP sites. As a whole, these results highlighted the AMF community assemblages’ responsivity to P but also showed that P is currently not a strong or prevalent ecological pressure exerted on the AMF community composition across the sampling area.

Conversely, our results suggest that contemporary AMF community assemblages may prevalently reflect life-history strategies and resilience of AMF communities, in response to mining history (i.e., land disturbance then ecosystem restoration), favoring Glomerales that behave as ruderals, able to rapidly respond to stressful conditions through their ability to disperse, colonize, and persist in soils and roots over time during reclamation of land-mined soils.

In conclusion, this work opens new opportunities to advance our in-depth understanding of the analysis of P’s impact on AMF communities. It also improves the knowledge of AMF diversity in natural P-rich soil, thus providing novel perspectives for the design of AMF inoculants to be used in combination with RP fertilization and the development of more sustainable agroecosystems.

## Figures and Tables

**Figure 1 microorganisms-10-02406-f001:**
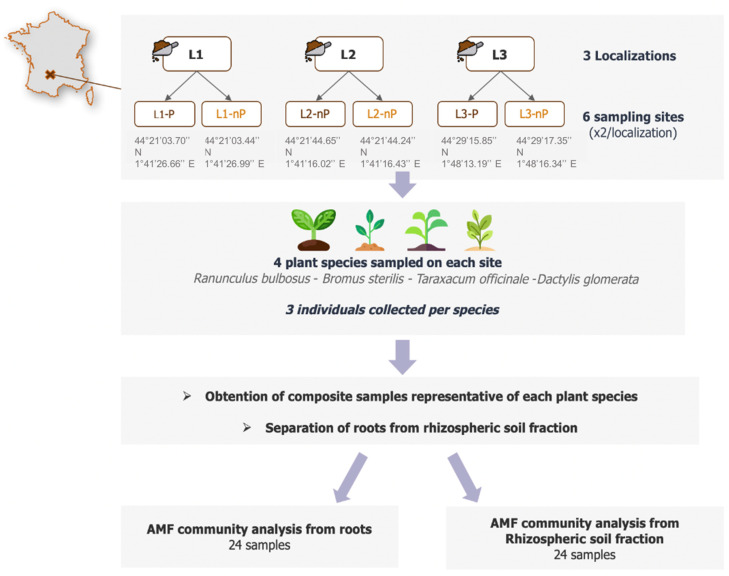
Sampling sites and experimental design.

**Figure 2 microorganisms-10-02406-f002:**
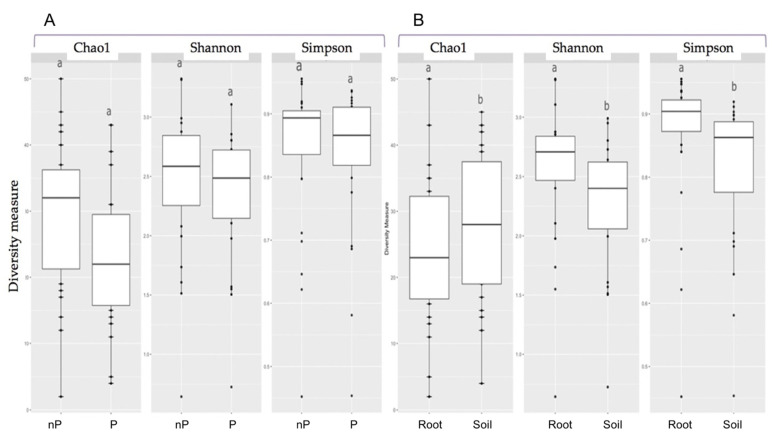
Richness (Chao 1 estimator) and diversity (Shannon and Simpson) indexes comparing AMF communities of (**A**) P versus nP soils, root and soil data being pooled; (**B**) root versus rhizospheric soil samples, P and nP site data being pooled. Within each graph, different letters denote significant differences using the Kruskal–Wallis nonparametric test (*p* < 0.05).

**Figure 3 microorganisms-10-02406-f003:**
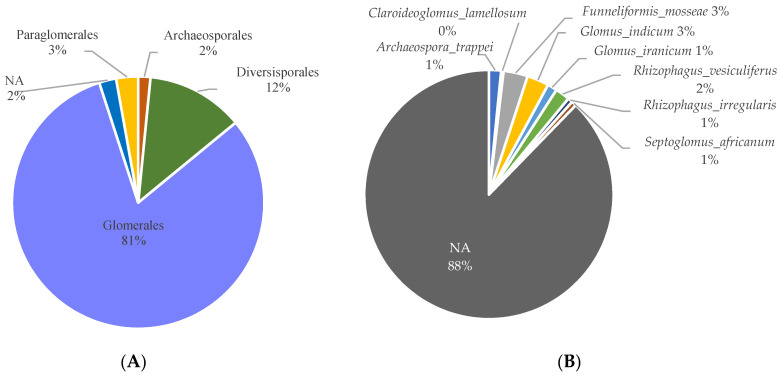
Taxonomic distribution of the 318 ASVs identified as Glomeromycota inside the entire mining area (**A**) at the order level, (**B**) at the species level, and (**C**) taxonomic distribution of the 318 ASVs at the family level inside each sample. “NA” category: ASV that did not obtain taxonomic assignment at the studied level.

**Figure 4 microorganisms-10-02406-f004:**
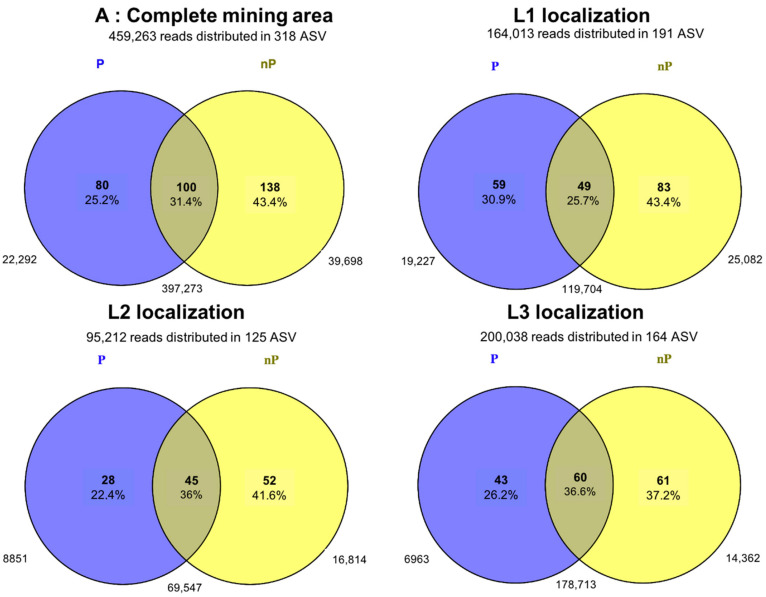
Venn diagrams showing the overlap of the AMF communities across P and nP sites, considering the entire mining area (A) or each localization individually (L1 localization, L2, and L3). Shared and specific ASV numbers and their relative abundance (in percent) are given inside the circles; the read numbers are given out of the circles.

**Figure 5 microorganisms-10-02406-f005:**
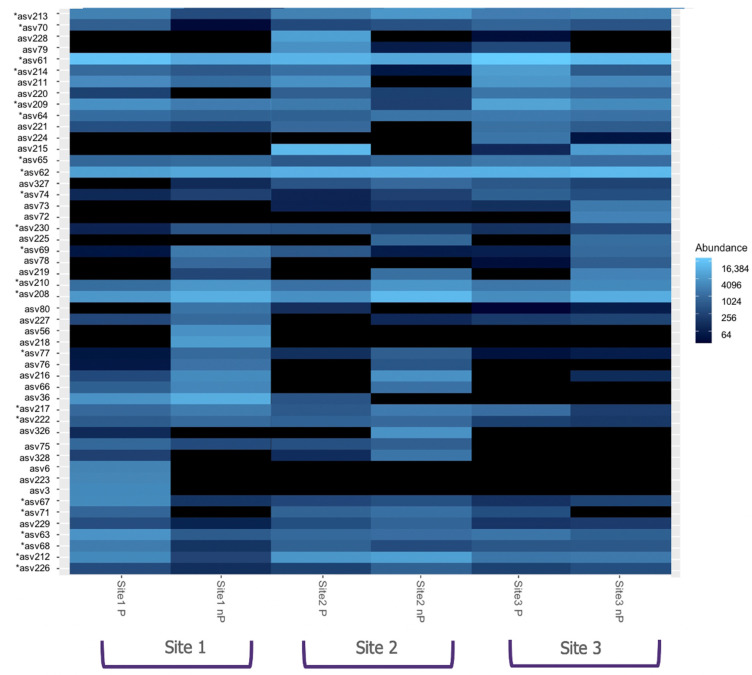
Heatmap representation showing the abundance of the 50 most abundant ASVs across all the sampling sites. The 50 most abundant ASVs include the 26 core ASVs shared by the six sampling sites *: Core ASVs shared by all sampling sites.

**Figure 6 microorganisms-10-02406-f006:**
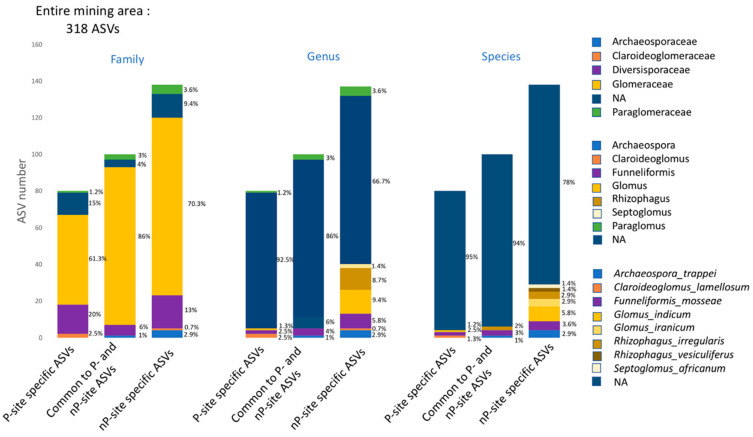
Distribution at the family, genus, and species level of the shared and specific ASVs identified in P and nP sites across the total mining area, NA: no-assigned ASVs.

**Table 1 microorganisms-10-02406-t001:** Mycorrhizal rates of the four plant species (*Ranunculus bulbosus*, *Bromus sterilis*, *Taraxacum officinale,* and *Dactylis glomerata*) grown on the six sites (L1-P and -nP, L2-P and -nP, and L3-P and -nP). Data based on 135 root fragments from three plants per species in each sampling site.

		Mycorrhizal Rates (%)					
Sites	Total P Concentration (mg kg^−1^)	*Ranunculus bulbosus*	*Taraxacum officinale*	*Dactylis glomerata*	*Bromus sterilis*	Average Per Site	Average P versus nP
L1-P	2860	38	40	33	51	41	
L2-P	13,928	20	27	42	18	27	42
L3-P	10,739	53	53	62	63	58	
L1-nP	1380	53	58	40	56	52	
L2-nP	1067	47	13	5	16	32	43
L3-nP	1496	58	44	33	42.	44	

## Data Availability

The original contributions presented in the study are publicly available.

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
