# Peer review of "Glomerales Dominate Arbuscular Mycorrhizal Fungal Communities Associated with Spontaneous Plants in Phosphate-Rich Soils of Former Rock Phosphate Mining Sites"

_microorganisms, 2022, doi:10.3390/microorganisms10122406_

Round 1

Reviewer 1 Report

The study by Ducousso-Détrez et al. intends to analyze the effect of P on arbuscular mycorrhizal fungi (AMF) diversity on roots and rhyzospheric soil. They used a metagenomics approach to compare AMF diversity on a mine site with contrasting P concentration. From the reading of the manuscript, the experimental design seems ok and the methodology adequate. However, the manuscript needs major improvements, mostly on Results and Discussion. The results are not clearly presented (Figures) and only high taxonomic levels are mentioned (family); no genus or species of AMF? Many paragraphs are confusing and need major improvement and focus. In addition to the observations made on the manuscript, authors should improve the following aspects:

The abstract needs to include some conclusions from the study.

The study area and experimental design description must be improved – why were those 3 geographical locations chosen? What’s the major differences between them? An explanation for the high P content in one site and low P in the other must be given. The aims of the study are not clearly defined.

For sequencing of AMF DNA, authors used the primer pair developed by Lee et al. (2008), AML1 and AML2, which is not compatible with Illumina sequencing, and then performed a nested PCR using an in-house designed primer pair. Are these in-house primers published in any publication? I don’t understand why authors didn’t use the primers developed to sequence AMF DNA, developed by Morgan and Egerton-Warburton (2017), specifically for Illumina sequencing (NS31/AML2)? Furthermore, the primers AML1/AML2 are known to exclude some AMF taxa, such as the Archaeosporaceae and Paraglomeraceae (Morgan and Egerton-Warburton, 2017). 

In the materials and Methods section, the bioinformatics analysis description is very poor. There are several items missing: read quality criteria, alignment of reads against some AMF database; alignment criteria. Also, the statistical analysis is not clear: did the authors calculated taxa abundance? Did they compare taxa abundance or alpha diversity between sites? From the reading of the materials and methods section, one cannot understand what were the comparisons made and what was being compared?

The results section needs major improvements. In addition to the observations made on the manuscript, authors should provide the following:

1 – In 3.2, a Table/graphic showing the number (or %) of ASVs (silva database and phylogenetics) identified at each experimental condition.

2 – A table showing all the ASVs identified must be provided as a supplementary file.

Regarding the Results, the manuscript is disorganized, with most of the figures not clearly showing the results described on the text. Most of the important results are on the supplementary files (table S2, table S4, table S5).

Discussion must be revised. It needs to be more focused – what were the major findings? How these can advance knowledge on this area of research. Was the phylognetics approach usefull for taxa identification? Needs major English editing.

Author Response

Reviewer 1:

  • Comment 1: The abstract needs to include some conclusion from the study.

Response: We thank the reviewer for this comment. As you can see in this revision, the abstract was re-written and we included new sentences that summarize the results of the study as well as its main conclusions.

Arbuscular mycorrhizal fungi (AMF) are key drivers of soil functioning. They interact with multiple soil parameters, notably phosphorus (P). In this work, AMF communities of native plants, grown spontaneously on former mining sites, enriched (P-sites) or not with P (nP-sites) by mining cuttings of rock phosphate (RP), were studied. No significant differences were observed in root mycorrhizal rates of the plants when comparing P and nP-sites. The assessment of AMF di-versity and community structure using Illumina MiSeq metabarcoding, targeting 18S rDNA in roots and rhizospheric soils, showed a total of 318 Amplicon Sequence Variants (ASVs) of Glomeromycota phylum. No significant difference in the diversity was found between P and nP-sites. Glomeraceae species were largely dominant and form a fungal core of 26 ASVs, persistent and abundant in all sites. In P-soils, eight ASVs were identified by indicator species analysis. A trend towards an increase in Diversisporaceae and Claroideoglomeraceae and a reduction in Paraglomeraceae and Glomeraceae were noticed. These results provide new insights on AMF ecology in former RP mining sites; they documented that P concentration is a driver of AMF community structure in soils enriched in RP for the long term, but also suggested influence of land disturbance, ecosystem self-restoration, and AMF life history strategies as drivers of AMF com-munity profiles.

Line 18: “surface” has been deleted

Line 24: “has been” replaced by “was”

Line 48: Authors should focus more on the knowledge of AMF for plant P nutrition.

Response: We rephrased the sentence and implemented it with information according to the reviewer’s suggestion: “In particular, AMF increase uptake of relatively immobile ions (such as phosphate ions) for their host plant due to an extensive extraradical hyphal network which grows beyond the zone of direct uptake by roots. By this way, it contributes for a larger soil prospection, and enables the mycorrhizal plants to access larger soil P resources. Therefore, AMF have a great importance to productivity and plant growth in most ecosystems (1, 18).

Line 71: “low P fertilizer inputs” replaced by “with a reduction in the use of chemical fertilizers”.

Line 72: Add some more information on rock phosphate. Why is it a good alternative to P chemical fertilizer?

Response: As suggested, precisions have been provided: “In particular, how AMF communities interact with rock phosphate has attracted considerable attention. Rock phosphate is a natural, low industrial cost phosphate input that is more ecologically and sustainably acceptable than chemical fertilizers.

Line 74: “of” replaced by “in shaping”

Line 75: “a controversial subject in the literature”: not clear. Explain and give references.

Response: As suggested, we rephrased the sentence and provided the related references:

“Thus, the impact of P levels on the occurrence or colonization of AMFs, as well as on the abundance and richness of AMF communities is regularly discussed (28, 39).”

This point was the subject of a paragraph developed extensively in the discussion.

Line 79: “inside” replaced by “in” 

Line 79: “ore”: deleted

  • Comment 2: The study area and experimental design description must be improved – why were those 3 geographical locations chosen? What’s the major differences between them? An explanation for the high P content in one site and low P in the other must be given. The aims of the study are not clearly defined.

Response: As suggested, precisions have been provided:

 This area is characterized by numerous paleokarsts whose fillings are enriched in phosphorite (i.e. a phosphatic ore containing variable proportions of tricalcium phosphate). There, in the past, mining extraction was carried out in the form of open-pit mines (Figure S1a), and some excavated spoils were abandoned near the extraction points. Thus, each selected site corresponds to a former mining point (i.e. a site without enrichment with RP ores hereafter, referred hereafter to as nP-site) associated with its spoil area (i.e. a site with ore deposits due to mining exploitation process, referred to as P-site). Today, in this former mining area, the dominant landscapes are dry grasslands, punctually wooded (Figure S1).

The P concentrations being known to be more or less high in each karst phosphatiere fillings [46], three nP were therefore chosen, expecting significantly high P concentrations. The choice of locations was also constrained i) by the possibility of locating potential P and nP sites in close proximity to each other from the ground surface, and ii) by the necessary presence of the same set of species constituting the plant sampling. Thus, four native herbaceous mycotrophic species were sampled: Ranunculus bulbosus, Bromus sterilis, Taraxacum officinale and Dactylis glomerata, and three plants per species were collected.  From these 3 plants, the soil closely attached to roots (i.e. rhizospheric soil) was manually removed by gentle agitation and pooled into a single soil fraction. Then, the roots were washed with sterile water to get rid of remaining soil particles, and then pooled into one root fraction per species (Figure 1). Soil and roots were then respectively kept at 4°C and -20°C, until further DNA extraction. This was repeated for each plant species in each site, leading to 24 root samples (6 sites × 4 plant species), and 24 rhizospheric soil samples. Detailed soil properties for each site are available in supplementary data (Table S1).”

Line 90: “have been” replaced by “were”

Line 94: Species names have been italicised, as suggested

Line 97: Rhizospheric soil sampling process was described more precisely to describe rhizospheric soil collection, as requested.

Line 101: Sentence added: “Detailed soil properties for each site are available in supplementary data (Table S1).”

  • Comment 3: For sequencing of AMF DNA, authors used the primer pair developed by Lee et al. (2008), AML1 and AML2, which is not compatible with Illumina sequencing, and then performed a nested PCR using an in-house designed primer pair. Are these in-house primers published in any publication? I don’t understand why authors didn’t use the primers developed to sequence AMF DNA developed by Morgan and Egerton-Warburton (2017), specifically for Illumina sequencing (NS31/AML2)? Furthermore, the primers AML1/AML2 are known to exclude some AMF taxa, such as the Archaeosporaceae and Paraglomeraceae (Morgan and Egerton-Warburton, 2017).

Response:

The in-house primers have been previously used by Renaut et al., 2020, as indicated in line 153 in the first version of the manuscript, and successfully used by other authors since then. Stefani et al., 2020, who designed and successfully used this in-house pair of primers, should also be acknowledged.

These references have been added for more precisions, and to take into account the comments of reviewer 1 in line 149. This correction led to a change in the order of bibliographic references which was taken into account in the correction of the reference list.                          

 “The DNA of AMF was specifically targeted using Nested-PCR (Surecycler 8800, Agilent Technologies, Les Ulis, France), with a set of two primer pairs as previously published by [50].  Thus, the first-round PCR was performed using the AMF-discriminating primer pair AML1 (3’-ATCAACTTTCGATGGTAGGATAGA-5’) and AML2 (3’-GAACCCAAACACTTTGGTTTCC-5’), which generates amplicons of the small 18S subunit of the rRNA gene, of about 800bp in length [51,52]. The PCR conditions were as follows: initial denaturation at 94°C for 3 min., followed by 35 cycles at 94°C for 1 min., 45°C for 1 min., and 72°C for 1 min., and a final elongation step at 72°C for 5 min [13]. PCR reactions were performed in a reaction volume of 25 µL and reagents were as follows: 5 µL of Q5 (5X) reaction buffer, 0.25 µL of Q5® High-Fidelity DNA Polymerase (New England Biolabs France, Evry, France), 0.8 µL of each primer (0.4 µM), 1 µL of dNTPs (0.2 mM), 1 µL of DMSO, 1 µL of BSA (100 µg.mL-1), and 1 ng of DNA template [13, 51].”

Line 105: “for” replaced by “from”

Line 155: This is a very low temperature.

Response: As underlined by Reviewer 1, the publication of Lee et al. refers to a temperature of 50ºC. We agree that a temperature of 45°C might appear as rather low, however Raveau et al., 2021 [13] provided relevant results, while using this annealing temperature. Thus, a quotation of this reference was added in the text to be more precise.

Line 159: this is very low DNA amount only detectable by qPCR.

Response: Such amount were previously reported by both Raveau et al. (2021) and Renaut et al., 2020, along with relevant results.

Raveau et al., 2021 has been added :“[13]”.

Line 159: “All PCR reactions were performed in triplicate”: the sentence has been deleted and moved further to line 174 for clarity, as further suggested by the reviewer 1 in line 174.

Line 168: “in bold above”: deleted and references were added (Raveau et al., 2021 (13) and Renaut et al., 2020) (53).

Line 174: Not clear. Specify the samples which were sequenced.

Response: “The triplicate DNA extracts for each PCR sample were pooled together.” ​

was replaced by :

“From genomic DNA of root and soil samples, collected for each plant species harvested in each site, PCR reactions were performed in triplicates, before the replicates for each PCR product were pooled together.”

  • Comment 4: In the materials and Methods section, the bioinformatics analysis description is very poor. There are several items missing: read quality criteria, alignment of reads against some AMF database; alignment criteria.

Also, the statistical analysis is not clear: did the authors calculate taxa abundance? Did they compare taxa abundance or alpha diversity between sites? From the reading of the materials and methods section, one cannot understand what were the comparisons made and what was being compared?

Response: Please find here below, line by line, our responses to the reviewer's requests in line with comment 4.

Line 184: Not clear. 

Response: To enhance clarity and readability, this part was re-written.

Bioinformatic process and statistical analyses were operated in the R 4.0.2 software (R Core Team, 2019) environment. The DADA2 pipeline (v. 1.16) [54], an open-source program implemented in R package (https://benjjneb.github.io/dada2/tutorial.html), was used to process the sequencing data. Briefly, sequence reads were filtered and trimmed using optimised parameter settings as recommended. Sequence reads were de-replicated, de-noised, and merged using DADA2 default parameters. Then, sequences were aligned and categorize to infer amplicon sequence variants (ASV) grouping amplicon sequencing data by using 100% of sequence identity [55].

Callahan et al., 2017 [55] was added in the reference list.

Line 189: Describe the aim of the statistical analysis. Which samples are being compared?

Response: A sentence was added to clarify this point:

“To identify the influence of former RP inputs on AMF communities, we studied AMF communities across the six sampling sites, as well as inside root or rhizospheric soil, using as proxy richness, alpha diversity, distribution and abundance of taxa. Then, data relative to P sites were compared to those of nP sites, inside each location, or after pooling data from the 3 P sites on the one hand and the 3 nP sites on the other hand.” 

Line 208: Taxonomic assignment was moved to the section above, before alpha diversity, as suggested.

Line 209 to 215: Blast parameters at Silva database. Not clear. Needs editing.

Response: It should first be noted that the blast parameters at silva database are pre-defined while using the “assignTaxonomy()” command. A minimum bootstrap value of 80 was nonetheless selected to ensure assigning taxonomic level with an acceptable confidence level. As suggested, this section was re-written.

 “The taxonomic assignment of ASVs was performed following a previously established two-step approach (Stefani et al., 2020; Raveau et al., 2021). Firstly, the Silva v132 database formatted for DADA2 was used to assign ASVs from kingdom to genus (minimum bootstrap 80) with the assignTaxonomy() command and the blast parameters as constitutively formatted for DADA2 [59]. From the ASVs assigned as AMF with Silva database, affiliation of each ASV was assessed at the genus level by performing a BLAST analysis against the NCBI database and MARJAAM, a web-based database containing referenced Glomeromycota DNA sequence data. The ASVs identified as non-Glomeromycota at the phylum level, at the end of these two stages of assignment, were excluded from further analyses.”

Line 214: How many?

Response: 392 ASVs were inferred and identified as Glomeromycota. Among them, 74 ASVs were not assigned at phylum level after inferring the phylogenetic tree.

Line 217: Bibliography authors update for “Stefani et al. 2020”

Line 218 to 224: not clear.

Response: As suggested, this part was edited to enhance clarity.

In a second step, aiming at a refined taxonomic identification, multiple alignments, using consensus sequences [57] and the ASVs identified as Glomeromycota, were first performed using the web portal Kalign (Lassmann et al., 2009) (http://msa.sbc.su.se/)[58]. A maximum-likelihood tree was then calculated using RAxML v8.2.10 (Stamatakis, 2014)[59] through the CIPRES web portal (Miller et al., 2011)[60]. Visualization of the output was finally obtained with the FigTree v1.4.4 program (Figure S3).

Line 239: How many individual plants from each species?

Response: This piece of information was present in table 1 caption, as follows:

 “Data based on 135 root fragments from three plants per species in each sampling site.”

Comment 5: The results section needs major improvements. In addition to the observations made on the manuscript, authors should provide the following

1 – In 3.2, a Table/graphic showing the number (or %) of ASVs (silva database and phylogenetic) identified at each experimental condition.

Response:

As suggested, a table was provided as supplementary material to complement previously given information. It can be found following this link: Table S3; https://drive.google.com/drive/folders/1E8DLOBk-wnGQTGW5iCnKP3K2Z1hvoHv7?usp=sharing.

2 – A table showing all the ASVs identified must be provided as a supplementary file.

Response:

As suggested, a table was provided as supplementary material. It can be found following this link: Table S4;  https://drive.google.com/drive/folders/1E8DLOBk-wnGQTGW5iCnKP3K2Z1hvoHv7?usp=sharing

  • Comment 6: Regarding the Results, the manuscript is disorganized, with most of the figures not clearly showing the results described on the text. Most of the important results are on the supplementary files (table S2, table S4, table S5).

Response:

Please find here below, line by line, the revisions that have been carried out in line with this comment.

Line 242: Provide a table with the number of reads in each treatment and plant species.

Response:

As suggested, a table S2 has been provided in the supplementary data section.

Line 244 to 246: Not clear. Justify why you think it's adequate.

Response:

This part was re-written.

“The rarefaction curves obtained after quality filtering reached the plateau, this way indicating that we had a good representation of the microbial community as most of the abundant species are represented and that sequencing depth effort was adequate to progress in further AMF community analysis (Figure S2).”

Line 247: Provide Tables (Supplementary material) with all ASVs identified as AMF by the Silva database and those identified de novo by the phylogenetics analysis.

Response:

Table S3 (Taxonomic distribution of the ASVs identified as AMF from the Silva database and from the phylogenetics analysis) was added in Supplementary material.

Line 249: The Table S3 citation was accordingly added

Line 250: “for” was replaced by “from”

Line 250: “and 459 263 reads” was deleted

Line 251: This suggests that the aim of the study was to compare soil with roots, which I think it's not the case. Re-formulate. Figure S2 does not seem to fit here.

Response: You are right, this formulation could be misleading. As such, “within the soil and root samples across the 6 study sites” has been replaced by “within P and nP-soils”.

Line 252: “(Figure S2)” has been suppressed and reference to the phylogenetic tree  (Figure S3) was moved : “Thus, a total of 318 ASVs (Figure S3) were kept for further comparisons of the AMF communities ”

Line 268: Not clear.

Response:

To enhance clarity, as suggested, the sentence “After the taxonomic and phylogenetic analyses.... experimental conditions” was suppressed.

Line 272: “Among all of them” replaced by “Among them”

Line 274: Since in this paragraph you are not analysing variations across samples, I don't see the point in Fig 3B at this stage.

Response: You are right, this quote was misplaced. “with variation across samples (Figure 3B)” was hence deleted from this location and reported at the end of the section.

Line 274 to 286: This is important results (a long paragraph describing identifications at the genus and species level). Insert 1 or 2 figures describing these results (%). Include those unidentified. Include the several experimental conditions.

Response:

One figure (Figure 3B) has been inserted to provide data (%) describing identifications at the species level with those unidentified as suggested. Figure 3C includes the several experimental conditions (root vs soil samples in each site).

The legend of Figure 3 has been modified as follows: “Figure 3. Taxonomic distribution of the 318 ASVs identified as Glomeromycota inside the entire mining area (A) at the order level, (B) at the species level and (C) Taxonomic distribution of the 318 ASVs at the family level inside each sample.

‘NA” category: ASV that did not obtain taxonomic assignment at the studied level”

In the text, a sentence was added at the end of the section 3.4:

“Regarding variations across the samples (Figure 3C), Glomeraceae were largely dominant in all samples, with Diversisporaceae and Paraglomeraceae mainly observed in soil”.

Line 296: delete. It's a repetition.

Response: You are right. As suggested, the sentence “and thus identified as core across the sites” was deleted.

Line 294 to 302: In my opinion these results should be in a figure and not as a supplementary table. It would benefit the manuscript. Include the results in the Figures above describing ASV classification for the experimental conditions.

Response: The data in Table S4 are indeed of interest to the manuscript.  However, it was chosen to leave them as supplementary data for more clarity in the reading of the article. The number of tables and figures would exceed the number classically allowed in the publication.

And it was chosen to keep them in table form for more completeness, than a bar chart or a presentation in pie chart would allow.

Line 298: Not clear.

Response:

To enhance clarity, “Three among them were prevalent in terms…. number, respectively”

has been replaced by:

“Among them, three were dominant, contributing to 57.1% of the read number of the 26 ASVs.”

Line 302: “rank” replaced by “level”

Line 308 to 313: Confusing

Response: To avoid misleading sentences,

“Thus, considering the complete mining area, we identified 100 ASVs that are shared by the two profiles, representing 31.4 % of the total ASV number, but 86.5 % of the total read number. In contrast, 138 ASVs (43.4 %), accumulating 8.6 % of reads, were identified as nP-site specific (hereafter referred to as nP-specific ASVs). Concomitantly, 80 ASVs (25.2 %) were P-specific, displaying 4.85 % of the reads (Figure 4).”

has been replaced by:

"Thus, considering the entire mining area, 100 ASVs shared by both profiles were identified, representing 31.4% of the total ASV number and 86.5% of the total read number. In contrast, 138 ASVs were identified as nP site-specific (i.e.  43.4% of ASVs and 8.6% of reads), while 80 ASVs were P-site specific (i.e. 25.2% of ASVs and 4.85% of reads (Figure 4)."

Line 315: There is no B in the figure

Response:

The figure 4 caption has been replaced by

“Figure 4: Venn diagrams showing the overlap of the AMF communities across P- and nP-sites, considering the entire mining area (A) or each localization individually (L1 localization, L2 and L3). Shared and specific ASV numbers and their relative abundance (in percent) are given inside the circles; the read numbers are given out of the circles.”

Line 319:

Response: As suggested by the reviewer, “consisting in lower richness…. ASV community.”

was replaced by

“with P-sites showing a lower number of specific ASVs, compared to non-P sites”.

Line 323: “identity”:  was deleted as suggested

Line 324: “inside the total AMF dataset” was deleted, as suggested.

Line 325: Not clear. Add some important results in the text based on the figure or move the figure to supplementary files.

Response:

As suggested, a sentence was added for concluding on figure 5: “In particular, the persistence and abundance of ASV 61, 62 and 208 in all sites was highlighted.”

Line 332: These are important results that identify which AMF taxa are specific to P and nP soils. A figure should be added to show these results.

Response:

Indicator species are often determined using an analysis between the species occurrence or abundance values from a set of sampled sites and the classification of the same sites into site groups, which may represent habitat types. Thus, indicator species analysis is the result of statistical calculations, i.e. the indicator values as a measure of specificity and fidelity of each ASV to the targeted clustering groups, i.e. P- vs nP-site group, as explained in the Discussion section, line 481). Such data are classically published in table form to avoid a consequent loss of information, in particular significance of the statistical analysis and the classification of the sampling sites into groups).

“Thus” was suppressed, as suggested

Line 335 to 337: Comparisons should focus on % and not on read number

Taxa corresponding to lower classification levels (genus/species) must be inserted in the figure. Re-formulate the paragraph.

Response:

Figure 6 was updated as suggested by reviewer 1: i.e. with % of ASV and not read; Distribution at the family, genus and species level was provided.

“Regarding the taxonomic classification, no Archaeosporaceae member was observed in P-soils, while Paraglomeraceae tend to be reduced in P-soils in terms of ASV (%) (Figure 6). The Diversisporaceae family tended to be relatively indifferent to the P content.”

has been replaced by

“Regarding the taxonomic classification, no Archaeosporaceae member was observed in P-soils. Paraglomeraceae and Glomeraceae tend to be reduced in P-soils in terms of ASV percentages (Figure 6). In contrast, Diversisporaceae and Claroideoglomeraceae families tended to have increased with the P content. Lower percentages of assigned ASV were recorded at genus and species levels in P-soils (Figure 6).”

Line 335: “in term of read number” was replaced by “in term of ASV number”

Line 334: “pattern” replaced by “classification”

Line 338: “a comparison inside the set of the three P-sites” replaced by “a comparison between the three P-sites”

Line 341: Confusing. Is there any different taxa between the 3 P sites?

Response: You are right, this could be misleading. To enhance clarity,

“In addition, a comparison inside the set of the three P-sites was performed (Figure S3A). The analysis revealed a low membership between them, characterised by a higher number of ASVs that are specific to one site, and by few overlapping ASVs, as shown by the Venn diagrams. Moreover, low-abundant ASVs mainly constituted the site-specific ASVs. The same trend occurred comparing the 3 nP-sites (Figure S3B).”

has been replaced by

"In addition, comparisons between the three P-sites were made. Different ASVs between the 3 P sites were identified, with a higher number of site-specific ASVs, and few shared ASVs, as shown in Venn diagrams (Figure S4A). The same trend occurred comparing the 3 nP-sites (Figure S4B).

Line 344: “Taxonomic distribution” replaced by “Taxonomic classification” as suggested.

  • Comment 7: Discussion must be revised. It needs to be more focused – what were the major findings? How these can advance knowledge on this area of research. Was the phylogenetic approach usefull for taxa identification?

Needs major English editing.

Response:

Few paragraphs of the Discussion have been revised as suggested by reviewer 1, also taking into account the remarks of Reviewer 2. In line with this comment, please find here below the revisions carried out, line by line.

Line 354 to 357: Not clear.

“Such contrasted conditions provided the opportunity to test to what extent community structure is responsive to a strong environmental P gradient, and to disentangle the potential role of P content as a driver of AMF communities.”

has been replaced by:

“In any case, such variations in P content provided the opportunity to examine the potential role of P content as a driver of AMF communities”.

Line 370: Bibliography authors update - Higo et al, 2020

Line 372: not clear

“Thus, AMF response to soil P could be related by concentration in soil”

was deleted and reformulated with the rewriting of the paragraph as suggested by the reviewer 1

Line 376: Bibliography authors update - Tang et al, 2016

Line 386 to 387: Not clear. Too long

“Taxonomic similarity at family level, prevalence of Glomeraceae and occurrence of a set of ASVs persisting across the P gradient: are these the consequences of site mining history?”

has been replaced by

"AMF community characteristics across sites: consequences of site mining history rather than P concentrations".

Line 388: deleted

“despite the large P gradient identified across the mining sites” was deleted as suggested

Line 393 to 395: Not clear. Delete “despite the sharp P gradient”

“In addition, we highlighted a small fraction of 26 ASVs corresponding to 63 % of the total read count, that are shared by all sampling sites despite the sharp P gradient.”

has been replaced by

“In addition, we highlighted 26 ASVs shared by all sampling sites, three of which being very abundant”.

Reviewer 2 Report

The manuscript "Glomerales dominate arbuscular mycorrhizal fungal communities associated with spontaneous plants in phosphate-rich soils of former rock phosphate mining sites" authored by: Amandine DUCOUSSO , Robin Raveau , Joël Fontaine , Mohamed Hijri , Anissa LOUNES - HADJ SAHRAOUI provided a very interesting insight into the arbuscular mycorrhizal fungal communities of P mining sites in France. The article is very well written, and very interestingly presented and I enjoyed reading it. I especially like the discussion on the impact of high soil P on the taxonomical structure of the communities. However, I have some suggestions that, in my opinion, are absent from the approach:

1. I would suggest describing the plant communities in sites where the samples were taken. The reader cannot really form an opinion on your work unless you provide what kind of ecosystems were you working in. To me, it is not clear what are the natural sites that are not treated with P ore depositions (pastures, fields?). The plant species that you have used for your observation are very common and certainly ruderal - this could be also the reason for having low diversity of AMF families and would explain the domination of ruderal AMF. Please, elaborate in the text.

2. If I understood correctly, the form of P that has been added was tricalcium phosphate. This form of soil P is considered the most insoluble and therefore practically unavailable to plants. The huge amount of total P in the soil is not surprising, but in measured soil pH values (neutral), P, in general, is considered to be highly unavailable. High amounts of clay would enhance this effect. Therefore, the large amounts of available P (Olson) in P sites are surprising to me. Do you consider that AMF or some other soil organisms may be factors of P release from its unavailable form? Please elaborate. I would also recommend supplying soil analysis results. Please elaborate/discuss this concerning your results.

Author Response

Reviewer 2:

  • Comment 1: I would suggest describing the plant communities in sites where the samples were taken. The reader cannot really form an opinion on your work unless you provide what kind of ecosystems where you are working in. To me, it is not clear what are the natural sites that are not treated with P ore depositions (pastures, fields?). The plant species that you have used for your observation are very common and certainly ruderal - this could be also the reason for having low diversity of AMF families and would explain the domination of ruderal AMF. Please, elaborate in the text.

Response :

As suggested by the reviewer, we have provided precisions about the ecosystem in the sampling sites. To our knowledge, no published exhaustive flora inventory is available for the mining area, but photographs have been provided as supplementary data (figure S1) to illustrate the ecosystem and the plant cover in the sampling sites.

In addition, section 2.1 (study area and experimental design) has been improved for more clarity regarding the differences between P and nP sites, as requested by the two reviewers as follows:

“This work was carried out in the region of the "Phosphatières du Quercy" (southern France; N 44.351827, E 1.691021). In this area, the dominant landscape are dry grasslands, punctually wooded (Figure S1). Also, numerous phosphatieres that are paleokarsts whose filling has the particularity of being enriched in phosphorite (i.e. a phosphatic ore containing variable proportions of tricalcium phosphate) are observed. There, in the past, mining was carried out following phosphate cavities and the extraction was carried out in the form of open-pit mines, as it can still be seen from the material remains (Figure S1). The spoil from the mining operations was then placed, closely, around the extraction points. Thus, each location chosen corresponds to a former mining point (i.e. a site without enrichment with RP ores hereafter, referred to as nP site) associated with its spoil area (i.e. a site with ore deposits due to mining exploitation process, referred to as P).

The P concentrations being known to be more or less high in each karst phosphatiere fillings [46], three nP were therefore chosen, expecting significantly high P concentrations. The choice of locations was also constrained i) by the possibility of locating potential P and nP sites in close proximity to each other from the ground surface, and ii) by the necessary presence of the same set of species constituting the plant sampling. Thus, four native herbaceous mycotrophic species were sampled : Ranunculus bulbosus, Bromus sterilis, Taraxacum officinale and Dactylis glomerata, and three plants per species were collected.  From these 3 plants, the soil closely attached to roots (i.e. rhizospheric soil) was manually removed by gentle agitation and pooled into a single soil fraction. Then, the roots were washed with sterile water to get rid of remaining soil particles, and then pooled into one root fraction per species (Figure 1). Soil and roots were then respectively kept at 4°C and -20°C, until further DNA extraction. This was repeated for each plant species in each site, leading to 24 root samples (6 sites × 4 plant species), and 24 rhizospheric soil samples. Detailed soil properties for each site are available in supplementary data (Table S1).”

  • Comment 2: If I understood correctly, the form of P that has been added was tricalcium phosphate. This form of soil P is considered the most insoluble and therefore practically unavailable to plants. The huge amount of total P in the soil is not surprising, but in measured soil pH values (neutral), P, in general, is considered to be highly unavailable. High amounts of clay would enhance this effect. Therefore, the large amounts of available P (Olson) in P sites are surprising to me. Do you consider that AMF or some other soil organisms may be factors of P release from its unavailable form? Please I would also recommend supplying soil analysis results. Please elaborate/discuss this concerning your results.

Response:

You are right, this result was unexpected. In that regard, and as suggested, we elaborated an additional paragraph, which has been added within the first paragraph of the Discussion section, about the large amounts of available P. However, we have made an effort to be concise, with respect to a previous request of reviewer 1 (suggesting that discussion needed to be more focused):

“Such high concentrations in soluble P content are scarce and unexpected. Indeed, in the phosphatieres, soils are enriched with apatite, a form of soil P highly insoluble, and with aluminous phosphate parageneses, clays and sands (Billaud, 1982). As orthophosphate ions act as chemical ligands for soil compounds, P is generally considered to be highly unavailable in such soils and in measured soil pH values (neutral). It is therefore appropriate to question the putative origin of such soluble P concentrations. Our work on the bacterial and fungal communities of the same sites revealed the presence of numerous taxa known to host phosphate solubilizing microorganisms (data not shown). Phosphate solubilizing bacteria have also successfully isolated. These microorganisms may be factors of P release from its unavailable form. However, this would imply that there is no feedback to regulate microbial P solubilizing metabolic activity in the presence of high phosphate substrate concentrations. But to the best of our knowledge, there is no data in the literature on this point.

In any case, such variations in total and soluble P content provided the opportunity to examine the potential role of P content as a driver of AMF communities.”

We also supplied a table with soil analysis results (Table S1) as suggested. Related quotation has been added at the end of paragraph 2.1.

Round 2

Reviewer 1 Report

I have no further comments and my only suggestion is for the authors to make a revision of English language.